# A Time Series Analysis Evaluating Antibiotic Prescription Rates in Long-Term Care during the COVID-19 Pandemic in Alberta and Ontario, Canada

**DOI:** 10.3390/antibiotics11081001

**Published:** 2022-07-26

**Authors:** Manon R. Haverkate, Derek R. Macfadden, Nick Daneman, Jenine Leal, Michael Otterstatter, Roshanak Mahdavi, Adam G. D’Souza, Elissa Rennert-May, Michael Silverman, Kevin L. Schwartz, Andrew M. Morris, Ariana Saatchi, David M. Patrick, Fawziah Marra

**Affiliations:** 1Faculty of Pharmaceutical Sciences, University of British Columbia, Vancouver, BC V6T 1Z3, Canada; manon.haverkate@ubc.ca (M.R.H.); ariana.saatchi@ubc.ca (A.S.); 2Ottawa Hospital Research Institute, University of Ottawa, Ottawa, ON K1Y 4E9, Canada; dmacfadden@toh.ca; 3ICES, Toronto, ON M4N 3M5, Canada; nick.daneman@sunnybrook.ca (N.D.); roshanak.mahdavi@gmail.com (R.M.); kevin.schwartz@oahpp.ca (K.L.S.); 4Public Health Ontario, Toronto, ON M5G 1M1, Canada; 5Division of Infectious Diseases, Department of Medicine, Sunnybrook Research Institute, Toronto, ON M4N 3M5, Canada; 6Department of Community Health Services and Department of Microbiology, Immunology and Infectious Diseases, Cumming School of Medicine, and O’Brien Institute of Public Health, University of Calgary, Calgary, AB T2N 1N4, Canada; jenine.leal@albertahealthservices.ca (J.L.); elissa.rennertmay@ucalgary.ca (E.R.-M.); 7Infection Prevention and Control, Alberta Health Services, Calgary, AB T1Y 6C2, Canada; 8British Columbia Centre for Disease Control, Vancouver, BC V5Z 4R4, Canada; michael.otterstatter@bccdc.ca (M.O.); david.patrick@bccdc.ca (D.M.P.); 9School of Population and Public Health, University of British Columbia, Vancouver, BC V6T 1Z3, Canada; 10Centre for Health Informatics, Cumming School of Medicine, University of Calgary, Calgary, AB T2N 1N4, Canada; adam.dsouza@albertahealthservices.ca; 11Data and Analytics, Alberta Health Services, Calgary, AB T1Y 6C2, Canada; 12Division of Infectious Diseases, Faculty of Medicine, University of Western Ontario, London, ON N6A 3K7, Canada; michael.silverman@sjhc.london.on.ca; 13Dalla Lana School of Public Health, University of Toronto, Toronto, ON M5S 1A1, Canada; 14Sinai Health, University Health Network, and University of Toronto, Toronto, ONM5S 1A1, Canada; andrew.morris@sinaihealth.ca

**Keywords:** antibiotic prescribing, long-term care, COVID-19, time series analysis, epidemiology

## Abstract

The COVID-19 pandemic affected access to care, and the associated public health measures influenced the transmission of other infectious diseases. The pandemic has dramatically changed antibiotic prescribing in the community. We aimed to determine the impact of the COVID-19 pandemic and the resulting control measures on oral antibiotic prescribing in long-term care facilities (LTCFs) in Alberta and Ontario, Canada using linked administrative data. Antibiotic prescription data were collected for LTCF residents 65 years and older in Alberta and Ontario from 1 January 2017 until 31 December 2020. Weekly prescription rates per 1000 residents, stratified by age, sex, antibiotic class, and selected individual agents, were calculated. Interrupted time series analyses using SARIMA models were performed to test for changes in antibiotic prescription rates after the start of the pandemic (1 March 2020). The average annual cohort size was 18,489 for Alberta and 96,614 for Ontario. A significant decrease in overall weekly prescription rates after the start of the pandemic compared to pre-pandemic was found in Alberta, but not in Ontario. Furthermore, a significant decrease in prescription rates was observed for antibiotics mainly used to treat respiratory tract infections: amoxicillin in both provinces (Alberta: −0.6 per 1000 LTCF residents decrease in weekly prescription rate, *p* = 0.006; Ontario: −0.8, *p* < 0.001); and doxycycline (−0.2, *p* = 0.005) and penicillin (−0.04, *p* = 0.014) in Ontario. In Ontario, azithromycin was prescribed at a significantly higher rate after the start of the pandemic (0.7 per 1000 LTCF residents increase in weekly prescription rate, *p* = 0.011). A decrease in prescription rates for antibiotics that are largely used to treat respiratory tract infections is in keeping with the lower observed rates for respiratory infections resulting from pandemic control measures. The results should be considered in the contexts of different LTCF systems and provincial public health responses to the pandemic.

## 1. Introduction

The coronavirus disease of 2019 (COVID-19) pandemic, caused by the severe acute respiratory syndrome coronavirus 2 (SARS-CoV-2) virus, has had a major impact on all facets of healthcare. Access to care changed, as more care was delivered from a distance through telehealth or videoconferencing. Furthermore, measures implemented to curb the spread of the virus, such as social distancing and wearing masks, influenced the transmission of other infectious diseases [1]. These changes have also impacted antibiotic prescribing.

In the United Kingdom, antibiotic prescribing rates in primary care were higher at the start of the pandemic (March 2020), but fell below expected rates between April and August 2020 [2]. Similar decreases in outpatient antibiotic prescriptions since the start of the pandemic have been found in Finland [3], the Netherlands [4], Spain [5], and Canada [6].

Within Canada, our team evaluated outpatient antibiotic prescribing in Ontario, and found a 31.2% relative reduction in total antibiotic prescriptions between March and December 2020 compared to the pre-pandemic period, and an even larger reduction in prescription rates for antibiotics indicated for respiratory tract infections (RTIs) [7]. Similar results were found in British Columbia, where overall and RTI-specific monthly antibiotic prescription rates declined significantly during April–July 2020 compared with the same months during pre-pandemic years [8].

Among persons living in long-term care facilities (LTCFs), antibiotic prescribing trends may be different than those found elsewhere. The LTCF population generally consists of persons of older age with multiple complex comorbidities, i.e., a vulnerable group more susceptible to infections and subsequent adverse outcomes [9]. Furthermore, resource limitations in this setting can have a major impact on the possibilities for infection prevention and control. For example, limited availability of personal protective equipment, staff shortages, and shared facilities are known to have facilitated the spread of SARS-CoV-2 in LTCFs [9,10]. Similar to other countries, LTCFs in Canada were hardest hit by COVID-19, with almost 80% of all COVID-19 deaths being LTCF residents [11]. Yet, patterns of antibiotic prescribing in LTCF during the COVID-19 pandemic remain largely unknown. The overarching objective of this study was to describe and understand oral antibiotic prescribing in LTCFs in two provinces in Canada (Alberta and Ontario) during the COVID-19 pandemic and the resulting control measures, using a novel approach of linked administrative data.

## 2. Results

Across the study years, the average annual cohort size was 18,489 for Alberta and 96,614 for Ontario (Table 1). The age distribution was similar for both provinces, with the majority of residents being over 85 years of age. In both provinces, about two thirds of the residents were female. The total number of yearly prescriptions dispensed in Alberta ranged from 11,510 to 12,828, and in Ontario, it ranged from 102,116 to 125,567. The majority of prescriptions were administered to individuals 80 years of age and older. An annual average of 30.5% of LTCF residents in Alberta and 51.9% of residents in Ontario received at least one antibiotic prescription. Beta-lactam antibiotics (ATC class J01C and J01D) and quinolones (ATC class J01M) were prescribed most often in both provinces, accounting for, on average, 67.2% of all prescriptions among LTCF residents in Alberta, and 65.1% in Ontario.

The overall trend in antibiotic prescription rates in the two provinces can be found in Figure 1. The prescription rates in Ontario were higher than in Alberta. A steady decline in weekly antibiotic prescription rates can be seen in Ontario pre-pandemic, with rates decreasing from 33.5 prescriptions per 1000 LTCF residents in 2017 to 31.0 per 1000 residents in 2019; this decreasing trend continued into 2020, with 28.7 prescriptions per 1000 residents. In contrast, a slight increase is visible in Alberta between 2017 and 2019, with an average of 17.5 prescriptions per 1000 LTCF residents in 2017 to 18.8 per 1000 residents in 2019, followed by a decrease in 2020 to 18.1 prescriptions per 1000 residents.

The decrease in average prescription rates in both provinces during the pandemic period (March–December 2020) compared to the same months in 2019 is shown in Table 2. The province of Alberta had 7.6% less prescriptions dispensed, and Ontario had 8.5% less prescriptions dispensed. Graphs showing the trend in prescription rates stratified by sex and ATC class are shown in Figure 2 and Figure 3, respectively.

To evaluate the impact of the pandemic on antibiotic prescriptions, while taking into account the trends in antibiotic prescribing pre-pandemic, we conducted an interrupted time series analysis (Table 3). The reported changes in weekly prescription rates in this Results section refer to step changes. A significant decrease was found in the overall weekly prescription rate after the start of the pandemic compared to pre-pandemic in Alberta (−3.9 per 1000 LTCF residents decrease in weekly prescription rate, 95% confidence interval (CI): −6.9–−1.0), but no significant change was seen in Ontario (−1.5 per 1000 LTCF residents decrease in weekly prescription rate, 95% CI: −4.0–1.0). More detailed analyses of demographic factors revealed a significant decrease in the weekly prescription rate in females in Alberta (−3.8 per 1000 LTCF residents decrease in weekly prescription rate, 95% CI: −7.1–−0.6), but not in males (−2.4 per 1000 LTCF residents decrease in weekly prescription rate, 95% CI: −5.6–0.7); no sex-related changed in the antibiotic prescription rate was seen in Ontario (−1.7 per 1000 LTCF residents (95% CI: −4.5–1.0) and −1.3 per 1000 LTCF residents (95% CI: −3.8–1.2) decrease in weekly prescription rate for females and males, respectively). There were no significant changes in the weekly prescription rate for the different age categories in either province, with the exception of 75−79-year-old residents in Ontario (−3.7 per 1000 LTCF residents decrease in weekly prescription rate, 95% CI: −6.6–−0.8).

When evaluating antibiotic prescription rates pre-pandemic versus pandemic, we saw changes in the prescribing of RTI-related antibiotics (Table 3). For example, in both provinces, significant decreases in the weekly prescription rate were seen within the class of beta-lactams/penicillins (J01C; Alberta: −1.3 per 1000 LTCF residents decrease in weekly prescription rate, 95% CI: −2.3–−0.4; Ontario: −1.2 per 1000 LTCF residents decrease in weekly prescription rate, 95% CI: −1.9–−0.5) and tetracyclines in Ontario (J01A; −0.2 per 1000 LTCF residents decrease in weekly prescription rates, 95% CI: −0.4–−0.04). The decrease seen within the class of penicillins was primarily related to the decrease in the use of amoxicillin in both provinces (Alberta: −0.6 per 1000 LTCF residents decrease in weekly prescription rate, 95% CI: −1.1–−0.2; Ontario: −0.8 per 1000 LTCF residents decrease in weekly prescription rate, 95% CI: −1.1–−0.5), and penicillin V in Ontario (−0.04 per 1000 LTCF residents decrease in weekly prescription rate, 95% CI: −0.06–−0.007), whereas the decrease related to the class of tetracyclines was related to decreases in the prescribing of doxycycline in Ontario (−0.2 per 1000 LTCF residents decrease in weekly prescription rate, 95% CI: −0.4–−0.06). Although we did not see an overall increase within the class of macrolides/lincosamides/streptogramins (J01F), we did see an increase in the prescribing of azithromycin, with a significantly higher rate of prescribing after the start of the pandemic compared to pre-pandemic years—although this was only seen in Ontario (0.7 per 1000 LTCF residents increase in weekly prescription rate, 95% CI: 0.2–1.2). Furthermore, a decrease was noted in the prescribing of oral cephalosporins (J01D) in both provinces, a class containing drugs used for the treatment of both RTIs as well as SSTIs (−0.8 per 1000 LTCF residents decrease in weekly prescription rate in both provinces; 95% CI: −1.5–−0.2 and −1.2–−0.3 for Alberta and Ontario, respectively).

No changes were seen in the prescription rates for drugs used to treat urinary tract infections (e.g., nitrofurantoin, Fosfomycin, and trimethoprim-sulfamethoxazole). For drugs used to treat SSTIs, a significant decrease in the prescription rate for cephalexin was found in both provinces (Alberta: −0.6 per 1000 LTCF residents decrease in weekly prescription rate, 95% CI: −1.1–−0.08; Ontario: −0.4 per 1000 LTCF residents decrease in weekly prescription rate, 95% CI: −0.7–−0.03).

Several sensitivity analyses were conducted to assess the robustness of our results (Appendix A). No major changes in the results were observed when models were adapted for the different sensitivity analyses. All significant step changes from the primary analyses were also found to be significant in at least two out of three sensitivity analyses, indicating the robustness of our results. When the gradual increase of the step function was modeled in 2 weeks instead of 3, or when a sudden increase in the step function was modeled (e.g., sensitivity analysis 1 and 2, respectively), only slight differences were seen in the results of the ITS analysis in the subgroups, i.e., some step or slope changes switched from significant to not significant or vice versa. However, they were approaching significance (close to *p* = 0.05) in the main analysis. In sensitivity analysis 3, where no slope change was incorporated, a higher number of significant results was found compared to the main analysis. This is probably due to the fact that incorporating a slope function might hide some of the step change.

## 3. Discussion

We evaluated the impact of the COVID-19 pandemic on oral antibiotic prescription rates in LTCFs in Canada, and demonstrated that overall antibiotic prescription rates significantly decreased in Alberta, but not Ontario, in the pandemic period compared to pre-pandemic. This difference may be related to the fact that the antibiotic prescription rate was already declining in Ontario pre-pandemic. Changes in the types of antibiotics prescribed during the pandemic were seen in both provinces. Antibiotics mainly used to treat RTIs, such as amoxicillin and doxycycline, were prescribed at a lower rate after the start of the COVID-19 pandemic, likely due to the public health measures that decreased the transmission of non-COVID related RTIs [12]. However, the proportion of LTC residents who received a visit from a physician dropped dramatically during the first wave of the pandemic, and the available data suggest that in-person visits from doctors were not entirely replaced with virtual visits [11]. This could partly explain the decrease in rates of respiratory-tract-related antibiotic prescribing.

A decrease in antibiotic prescription rates in the community following the start of the COVID-19 pandemic has been described previously on an international level [2,3,4,5,13], and within Canada [6]. Our results are similar to those of Knight et al., who found that in the outpatient setting in Canada, antibiotics mainly used for RTIs were prescribed at a much lower rate in the early months of the COVID-19 pandemic, which did not recover later on [6]. Of note, we saw an increased rate of azithromycin prescribing in Ontario, which may be due to the fact that this antibiotic was investigated as a possible treatment for COVID-19 early in the pandemic [14]. We did not see this antibiotic being used in increased quantities in Alberta, and this may be related to the fact that the pandemic wave began at a slightly later time in the latter province, and policies were already in place to only use azithromycin as COVID-19-related treatment within a clinical trial setting.

Studies evaluating the impact of the COVID-19 pandemic on antibiotic prescriptions, specifically in LTCFs, are still scarce. In the United States, a large study including 12% of US nursing homes showed a decrease in overall antibiotic prescribing [14]. However, this also included non-oral antibiotics. They also found significant decreases in the prescribing prevalence for amoxicillin, levofloxacin, cefuroxime, cephalexin, and trimethoprim-sulfamethoxazole, which is in line with our results. Contrary to our findings, prescribing for doxycycline was increased in their study. However, as they directly compared rates from 2020 to the same month in 2019, they omitted possible pre-existing trends and long-term seasonality. Another large study in Ontario, including all nursing homes in the province, found no significant changes in antibiotic prescription rates among nursing home residents after start of the pandemic, using interrupted time series analyses [15]. Although this is in line with our results for Ontario, they included both oral and non-oral antibiotics, and only looked at overall prescription rates.

As in many other countries, LTCFs were disproportionately affected by COVID-19 in Canada, with many infections and outbreaks and a high mortality rate. Yet, there were large differences within Canada [11]. LTCF residents comprised a much larger proportion of the total COVID-19 cases in Ontario compared to Alberta. During the peak of the first wave, Ontario experienced the largest increase in excess deaths of Canada (+28%) compared to Alberta (+15%) [11]. Although we did not see a significant change in overall oral antibiotic prescription rates in Ontario, contrary to what we found in Alberta, the prescription rates of more ATC classes and individual antibiotics were affected in Ontario compared to Alberta. Our findings may be explained by a decrease in non-COVID-19-related pneumonia, but with substantial amounts of inappropriate antibiotic prescribing for LTCF residents with COVID-19, which may explain the lack of a decrease in the prescription rates of amoxicillin-clavulanic acid and the increase in azithromycin prescriptions.

It is unknown what the effects will be of the observed changes in antibiotic prescription rates. In the short term, no increase in complications due to common bacterial infections was detected in Sweden, despite the decline in dispensed antibiotic prescriptions in a country with already low levels of prescribing [16]. In the longer term, the impact of the pandemic and the resulting changes in antibiotic prescription rates on antimicrobial resistance is yet to be determined [17,18,19]. Monitoring antibiotic prescription rates is of utmost importance to inform prescribing practices, and to provide insight into opportunities for antimicrobial stewardship. Knowing how antibiotic prescribing has changed during the COVID-19 pandemic can provide intervention targets and lead to a better understanding of the possible short- and long-term effects.

Retrospective studies using administrative data have inherent limitations. As the data are not specifically collected for research purposes, there is a possibility of several biases being introduced into the analyses. Furthermore, our rates are based on antibiotic dispensations, but levels of compliance of these medications are unknown. Moreover, we could not link antibiotic prescription to diagnoses, which poses a limitation to assessing the appropriateness of the prescription, as well as the indication they were used for. Our study design also means the causality of the association between the COVID-19 pandemic and changes in antibiotic prescription rates cannot be established with the methods used; other factors, such as changes in care (in person versus virtual visits), could also have caused these results in part. Future studies by our team will focus on the impact of telehealth versus in-person delivery of care on antibiotic prescribing. A further limitation to our study is the short time frame of available data, which was until 31 December 2020. As such, the impact of the different waves of the COVID-19 pandemic could not be assessed. Even though we have data on the complete population of LTCF residents in both provinces, p-values close to 0.05 should be interpreted with caution. In addition, data sources and database management are different between Alberta and Ontario; hence, rates may not be directly comparable between the provinces. For example, in Alberta, there is no comprehensive list of LTCFs with internal pharmacies that do not submit data to the provincial dispensations database. Therefore, there is a possibility that the denominators are inflated. However, trends would not be affected by this, and, thus, the comparison of trends, both pre-pandemic and after the start of the pandemic, is highly informative in assessing differences between the provinces regarding the impact of the pandemic. Importantly, differences between the provinces should be seen in the light of the differences in LTCF systems and responses to the pandemic.

Several strengths of this study are notable. We assessed prescription rates in detail for LTCF residents in two Canadian provinces, capturing the total population of LTCF residents in both provinces. Looking into subgroups of prescribing revealed trends that could not be seen when looking at overall prescription rates alone. Moreover, the statistical methodology (interrupted time series analysis using SARIMA) has several advantages. It is much more accurate and informative than comparing average prescription rates between pandemic versus pre-pandemic periods, as it counts for a variety of pre-existing trends. This is crucial for antibiotics, where stewardship programs have been in place for years, aiming to reduce prescribing [20,21,22,23]. Next to that, it takes seasonality into account, which is an important factor, especially for respiratory diseases and their associated antibiotic prescriptions. Besides accounting for seasonality and autocorrelation, we did not take other confounders into account in our models. However, since our study period is relatively short, we assume that there will not have been relevant changes in other important factors in this time period. Furthermore, our sensitivity analyses indicated the consistency and robustness of our results.

## 4. Materials and Methods

### 4.1. Study Design

A population-based retrospective cohort study was performed. The study cohort consisted of the total population of residents in LTCF in Alberta (AB) and Ontario (ON) within the study period. Only persons aged 65 and older were included. The ownership of LTCFs in Canada can be either private (profit or non-for-profit) or public, and differences exist between the provinces [24]. Data from all LTCFs in both provinces were included in this study. The study period was divided into the pre-pandemic period: 1 January 2017 to 29 February 2020 (*n* = 165 weeks); and the pandemic period: March 1, 2020 to 31 December 2020 (*n* = 44 weeks).

### 4.2. Data Sources

In Alberta, prescriptions dispensed by community pharmacies were captured by the Pharmaceutical Information Network (PIN) database. Denominator data were taken from the Alberta Continuing Care Information System (ACCIS), which contains admission, discharge, and transfer data to and from LTCFs in Alberta. ACCIS records were linked to the Alberta Hospital Discharge Abstract Database (DAD) to remove portions of LTCF stays during which the LTCF residents were admitted to acute care [25]. PIN records were linked to ACCIS records using the clients’ Unique Lifetime Identifier (ULI), a unique personal health number required to access most health care in Alberta. LTCFs that manage and report the dispensation of prescriptions internally and do not submit data to PIN were excluded from the analysis (*n* = 3), resulting in n = 185 facilities in Alberta that contributed data to the project over the course of the study period.

In Ontario, all data were accessed through ICES. ICES is an independent, non-profit research institute whose legal status under Ontario’s health information privacy law allows it to collect and analyze health care and demographic data, without consent, for health system evaluation and improvement. ICES datasets are linked using unique encoded identifiers and individual-level data analyzed at ICES. Virtually all residents have health insurance through the Ontario Health Insurance Plan (OHIP). Residents qualify for the Ontario Drug Benefit (ODB) program when they turn 65 years old. The ODB database is >99% accurate for identifying drug prescriptions for individuals 65 years of age and older [26]. Data on oral antibiotic prescriptions during the study period were extracted from this database. Denominator data from Ontario were taken from the Registered Persons Database (RPDB), which contains information on persons registered under OHIP, and who are eligible for the ODB. The residence of individuals within long-term care was determined using the Continuing Care Reporting System (CCRS) database [27]. Overall, *n* = 627 LTCFs in Ontario contributed data to the study.

### 4.3. Outcomes and Statistical Analyses

The primary outcome was the overall trend in the weekly rate of oral antibiotic use in LTCF residents during the pandemic, with comparison to historical trends in the prior three years, thereby correcting for seasonal effects. Only prescriptions with less than 30 days of supply were included, to exclude chronic use. Prescriptions of the same drug with a start date within three days of the expected end date of the previous prescription were combined as one prescription, to account for the fact that some prescriptions are dispensed in, for example, seven-day increments if prescribed for a longer time.

Prescription rates were calculated as prescriptions per 1000 LTCF residents per epidemiological week. Overall rates, rates by age and sex, and rates for the seven major antibiotic classes (as defined by the Anatomical Therapeutic Chemical (ATC) classification) were determined (see Appendix A) [28]. Furthermore, rates for nine individual antibiotics were calculated: amoxicillin, amoxicillin/clavulanic acid, azithromycin, cephalexin, clarithromycin, doxycycline, Fosfomycin, nitrofurantoin, and penicillin V. The most common indications for use of these individual agents (e.g., for respiratory tract infections, urinary tract infections, or skin and soft tissue infections) were adapted from Schwartz et al. [12], and can be found in Appendix A. To preserve subject anonymity, analyses were only performed if the number of dispensations was, on average, more than five per week.

Trends in antibiotic prescribing were explored with interrupted time series analyses using seasonal autoregressive integrated moving average (SARIMA) models, which account for seasonality [29]. The SARIMA model was structured as (p,d,q) × (P,D,Q)S, where p is the order of the autoregressive part of the model, d is the degree of non-seasonal differencing, and q is the order of the moving average part of the model. Furthermore, D is the degree of seasonal differencing, and P and Q are the autoregressive and moving average terms for the seasonal component. S is the seasonal period, which we set to 52, corresponding to weekly prescription rates. In line with the Box–Jenkins methodology [30], differencing was applied to satisfy the assumption of stationarity. The autocorrelation function (ACF) and partial autocorrelation function (PACF) plots were assessed to identify appropriate (p,q) and (P,Q) values. The best fitting model was determined through a stepwise search using an automated algorithm (auto.arima in R), which selects the model with the lowest AIC (Akaike information criterion), balancing model fit with a parsimonious number of model parameters.

To test for a change in the level and trend of prescribing after the onset of the pandemic, we included a step and a slope intervention term in the model. A step function indicates a sudden, sustained change, where the prescription rates are shifted either up or down by a given value directly following the start of the COVID-19 pandemic. The slope intervention function indicates a change in trend (either up or down) after the start of the pandemic [29]. For the main analysis, the step function was set to 0 before week 10 of 2020 (1 March), and smoothly increased to 1 in weeks 10−12 of 2020 (until 21 March) to capture the gradual imposition of public health measures in Canada in March 2020, a statistical technique that we have previously used to study antibiotic use in the community sector during the pandemic [8]. The slope function was set to 0 before week 10 of 2020 (1 March), and increased by 1 each week, starting in week 10 of 2020. Three sensitivity analyses were performed, in which: (1) a gradual increase of the step function was modeled in 2 weeks (week 10 and 11 of 2020); (2) a sudden increase of the step function was modeled (where the function was set to 0 before week 10 of 2020, and to 1 from week 10 of 2020); and (3) the step function identical to the main analysis was modeled, but without a slope function.

Time series data from both provinces were analyzed at the aggregate (population) level. All analyses were performed in R version 4.1.2 using the forecast package.

## 5. Conclusions

In conclusion, using linked administrative data, we found changes in the antibiotic prescribing in LTCFs after the start of the COVID-19 pandemic compared to pre-pandemic years in both Alberta and Ontario. A decrease in the overall antibiotic prescription rates in Alberta was seen after the start of the pandemic. Additionally, antibiotics largely used to treat RTIs were prescribed at a lower rate after the start of the pandemic in both provinces, which is in keeping with observed lower rates of overall respiratory infections associated with pandemic control measures. Furthermore, in LTCFs in Ontario, an increase was seen in prescription rates for azithromycin, a drug briefly investigated in the treatment of COVID-19. Our results suggest that the short- and long-term changes in antibiotic prescription rates and resistance, as well as the impact of subsequent waves of the COVID-19 pandemic, are important areas for further work.

## Figures and Tables

**Figure 1 antibiotics-11-01001-f001:**
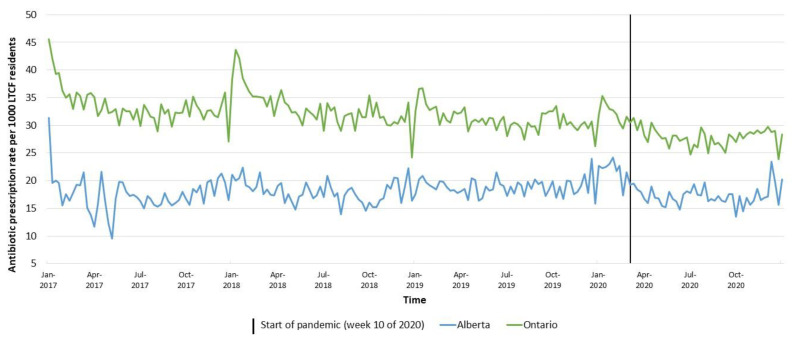
Overall oral antibiotic prescription rates per 1000 LTCF residents in Alberta and Ontario, Canada, 2017–2020. Footnote: Low rates at the end of a calendar year and high rates at the beginning of a calendar year represent administrative artifacts (e.g., less physician visits or less registration over the holidays, and more at the start of the new year).

**Figure 2 antibiotics-11-01001-f002:**
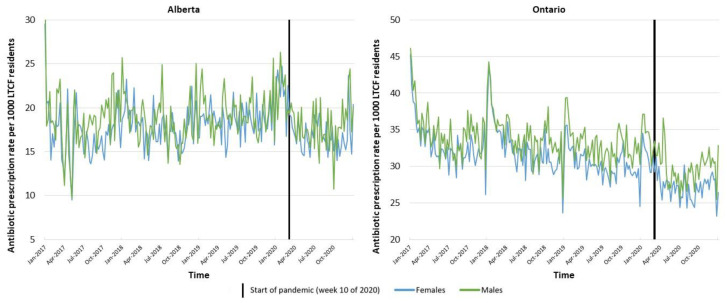
Oral antibiotic prescription rates per 1000 LTCF residents, Canada, 2017–2020, stratified by sex.N.B. Low rates at the end of a calendar year and high rates at the beginning of a calendar year represent administrative artifacts.

**Figure 3 antibiotics-11-01001-f003:**
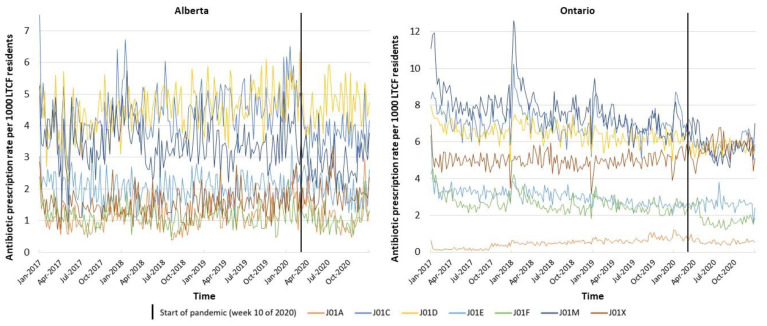
Oral antibiotic prescription rates per 1000 LTCF residents in Canada, 2017–2020, stratified by ATC (Anatomical Therapeutic Chemical) class. N.B. Low rates at the end of a calendar year and high rates at the beginning of a calendar year represent administrative artifacts.

**Table 1 antibiotics-11-01001-t001:** Yearly baseline characteristics of long-term care facility residents in Alberta and Ontario, Canada, 2017–2020.

		Alberta				Ontario			
		2017	2018	2019	2020	2017	2018	2019	2020
**Total Number of Residents**		18,137	18,605	18,962	18,253	98,815	99,842	99,332	88,468
**Age [n (%)]**	**65–69**	1039 (5.7%)	1055 (5.7%)	1136 (6.0%)	1131 (6.2%)	5197 (5.3%)	5315 (5.3%)	5350 (5.4%)	4613 (5.2%)
	**70–74**	1416 (7.8%)	1554 (8.4%)	1667 (8.8%)	1652 (9.1%)	7335 (7.4%)	7558 (7.6%)	7872 (7.9%)	7123 (8.1%)
	**75–79**	2132 (11.8%)	2270 (12.2%)	2341 (12.3%)	2293 (12.6%)	11,324 (11.5%)	11,710 (11.7%)	11,756 (11.8%)	10,546 (11.9%)
	**80–84**	3342 (18.4%)	3436 (18.5%)	3465 (18.3%)	3228 (17.7%)	18,432 (18.7%)	18,535 (18.6%)	18,258 (18.4%)	15,796 (17.9%)
	**85–89**	4339 (23.9%)	4395 (23.6%)	4348 (22.9%)	4120 (22.6%)	25,153 (25.5%)	24,924 (25.0%)	24,364 (24.5%)	21,213 (24.0%)
	**90+**	5869 (32.4%)	5895 (31.7%)	6005 (31.7%)	5829 (31.9%)	31,374 (31.8%)	31,800 (31.9%)	31,732 (31.9%)	29,177 (33.0%)
**Sex [n (%)]**	**Male**	6593 (36.4%)	6889 (37.0%)	7020 (37.0%)	6805 (37.3%)	30,996 (31.4%)	31,417 (31.5%)	31,641 (31.9%)	28,128 (31.8%)
	**Female**	11,544 (63.6%)	11,716 (63.0%)	11,942 (63.0%)	11,448 (62.7%)	67,819 (68.6%)	68,425 (68.5%)	67,691 (68.1%)	60,340 (68.2%)
**Total Number of LTCFs**		175	180	181	183	626	626	624	624
**LTCF Bed Count [n (%)] ***	**0–50**	80 (45.7%)	84 (46.7%)	80 (44.2%)	80 (43.7%)	64 (10.2%)	64(10.2%)	64 (10.3%)	64 (10.3%)
	**51–100**	43 (24.6%)	46 (25.6%)	49 (27.1%)	50 (27.3%)	210 (33.5%)	210 (33.5%)	209 (33.5%)	209 (33.5%)
	**101–150**	25 (14.3%)	23 (12.8%)	23 (12.7%)	24 (13.1%)	155 (24.8%)	155 (24.8%)	155 (24.8%)	155 (24.8%)
	**151–200**	10 (5.7%)	10 (5.6%)	12 (6.6%)	12 (6.6%)	124 (19.8%)	124 (19.8%)	124 (19.9%)	124 (19.9%)
	**>200**	17 (9.7%)	17 (9.4%)	17 (9.4%)	17 (9.3%)	73 (11.7%)	73 (11.7%)	72 (11.5%)	72 (11.5%)
**Total Number of Antibiotic Prescriptions**		11,510	11,920	12,828	12,337	125,567	123,841	116,542	102,116
**Percent of Residents with at Least One Antibiotic Prescription**		30.0%	29.9%	30.9%	31.3%	53.6%	52.9%	50.9%	49.8%
**Antibiotic Prescriptions by Age [n (%)]**	**65–69**	626 (5.4%)	630 (5.3%)	693 (5.4%)	784 (6.4%)	5871 (4.7%)	6454 (5.2%)	6218 (5.3%)	5703 (5.6%)
	**70–74**	917 (8.0%)	991 (8.3%)	1097 (8.6%)	1185 (9.6%)	9244 (7.4%)	9123 (7.4%)	9059 (7.8%)	8314 (8.1%)
	**75–79**	1249 (10.9%)	1472 (12.3%)	1588 (12.4%)	1468 (11.9%)	13,869 (11.0%)	13,817 (11.2%)	13,138 (11.3%)	11,699 (11.5%)
	**80–84**	2044 (17.8%)	2140 (18.0%)	2367 (18.5%)	2145 (17.4%)	22,917 (18.3%)	22,697 (18.3%)	21,001 (18.0%)	17,977 (17.6%)
	**85–89**	2833 (24.6%)	2816 (23.6%)	3010 (23.5%)	2680 (21.7%)	32,464 (25.9%)	31,697 (25.6%)	29,048 (24.9%)	24,624 (24.1%)
	**90+**	3841 (33.4%)	3871 (32.5%)	4073 (31.8%)	4075 (33.0%)	41,202 (32.8%)	40,053 (32.3%)	38,078 (32.7%)	33,799 (33.1%)
**Antibiotic Prescriptions by Sex [n (%)]**	**Male**	4165 (36.2%)	4198 (35.2%)	4612 (36.0%)	4498 (36.5%)	38,514 (30.7%)	38,233 (30.9%)	37,125 (31.9%)	32,655 (32.0%)
	**Female**	7345 (63.8%)	7722 (64.8%)	8216 (64.0%)	7839 (63.5%)	87,053 (69.3%)	85,608 (69.1%)	79,417 (68.1%)	69,461 (68.0%)
**Antibiotic Prescriptions by ATC [n (%)] ^#^**	**J01A—Tetracyclines**	682 (5.9%)	714 (6.0%)	862 (6.7%)	849 (6.9%)	821 (0.7%)	1885 (1.5%)	2531 (2.2%)	2208 (2.2%)
	**J01C—Beta-lactams**	2731 (23.7%)	2919 (24.5%)	3144 (24.5%)	2735 (22.2%)	26,857 (21.4%)	26,676 (21.5%)	26,277 (22.5%)	21,937 (21.5%)
	**J01D—Other beta-lactams**	2693 (23.4%)	3011 (25.3%)	3246 (25.3%)	3255 (26.4%)	24,841 (19.8%)	24,897 (20.1%)	23,257 (20.0%)	20,664 (20.2%)
	**J01E—Sulfonamides and trimethoprim**	1355 (11.8%)	1289 (10.8%)	1395 (10.9%)	1413 (11.5%)	12,486 (9.9%)	11,784 (9.5%)	10,027 (8.6%)	9222 (9.0%)
	**J01F—Macrolides, lincosamides and streptogramins**	783 (6.8%)	717 (6.0%)	783 (6.1%)	775 (6.3%)	10,694 (8.5%)	9889 (8.0%)	9357 (8.0%)	6895 (6.8%)
	**J01M—Quinolones**	2296 (19.9%)	2303 (19.3%)	2272 (17.7%)	2042 (16.6%)	31,217 (24.9%)	30,260 (24.4%)	26,426 (22.7%)	21,404 (21.0%)
	**J01X—Other antibacterials**	970 (8.4%)	967 (8.1%)	1126 (8.8%)	1268 (10.3%)	18,651 (14.9%)	18,450 (14.9%)	18,667 (16.0%)	19,786 (19.4%)

LCTF: Long-term care facility. * Bed counts are as of 31 March of the given year for Alberta, and as of 1 January of the given year for Ontario. ^#^ See Appendix A for all antibiotics included. J01C includes penicillins; J01D includes oral cephalosporins; and J01X includes amongst other metronidazole, nitrofurantoin and fosfomycin.

**Table 2 antibiotics-11-01001-t002:** Average weekly oral antibiotic prescription rates per 1000 LTCF residents and relative change in prescription rate in 2020 versus 2019, in long-term care facilities in Alberta and Ontario, Canada.

	Average Weekly Antibiotic Prescription Rate per 1000 LTCF Residents				Relative Change 2020 vs. 2019	Relative Change Mar–Dec 2020 vs. Mar–Dec 2019
	2017	2018	2019	2020		
**Alberta**	17.5	18.0	18.8	18.1	−3.9%	−7.6%
**Ontario**	33.5	33.1	31.0	28.7	−7.6%	−8.5%

**Table 3 antibiotics-11-01001-t003:** Interrupted time series analysis showing the change in weekly oral antibiotic prescription rate per 1000 LTCF residents after March 2020 in long-term care facilities in Alberta and Ontario, Canada.

		Alberta							Ontario						
Prescription Rate Category		Model Para-Meters *	Step Change	95% CI	*p*-Value	Slope Change	95% CI	*p*-Value	Model Para-Meters *	Step Change	95% CI	*p*-Value	Slope Change	95% CI	*p*-Value
**Overall**		(0,1,2) (1,0,0)[52]	−3.9461	−6.9209–−0.9714	**0.009**	0.0246	−0.1046–0.1539	0.709	(0,1,1) (0,1,1)[52]	−1.4665	−3.9554–1.0223	0.248	0.0210	−0.0966–0.1386	0.726
**Sex**	**Females**	(1,1,1) (0,0,1)[52]	−3.8337	−7.0746–−0.5929	**0.020**	0.0133	−0.0949–0.1215	0.809	(0,1,1) (0,1,1)[52]	−1.7110	−4.4635–1.0416	0.223	0.0282	−0.1033–0.1597	0.674
	**Males**	(1,0,1) (1,0,0)[52]	−2.4288	−5.5929–0.7354	0.132	0.0483	−0.0643–0.1609	0.401	(0,1,1) (0,1,1)[52]	−1.3071	−3.8369–1.2227	0.311	0.0138	−0.0787–0.1064	0.770
**Age**	**65−69**	ARIMA (0,0,0)	−0.0336	−3.0056–2.9385	0.982	0.0487	−0.0603–0.1579	0.381	(1,0,2) (1,0,0)[52]	0.8050	−3.2837–4.8938	0.700	−0.1122	−0.2575–0.0331	0.130
	**70−74**	ARIMA (0,0,5)	−2.4935	−5.8617–0.8747	0.147	0.0937	−0.0293–0.2166	0.135	(0,1,1) (1,0,0)[52]	−2.0866	−4.8668–0.6935	0.141	−0.0021	−0.1056–0.1014	0.968
	**75−79**	ARIMA (0,1,1)	−2.0757	−5.3594–1.2080	0.215	−0.0872	−0.2022–0.0279	0.138	(0,1,2) (1,0,0)[52]	−3.7163	−6.6276–−0.8049	**0.012**	0.0152	−0.0916–0.1220	0.780
	**80−84**	(0,1,1) (1,0,0)[52]	−3.3379	−6.9202–0.2444	0.068	−0.0221	−0.1545–0.1104	0.744	(0,1,1) (0,1,1)[52]	−2.1184	−4.5632–0.3263	0.089	−0.0093	−0.1024–0.0837	0.844
	**85−89**	ARIMA (1,0,1)	−3.1022	−6.6355–0.4310	0.085	0.0492	−0.0783–0.1767	0.450	(0,1,1) (1,0,0)[52]	−1.2703	−5.3305–2.7899	0.540	0.0141	−0.2026–0.2308	0.898
	**90+**	(1,0,1) (1,0,0)[52]	−2.0731	−6.0552–1.9090	0.308	0.0432	−0.0950–0.1814	0.540	(0,1,1) (0,1,1)[52]	−1.0053	−4.0054–1.9948	0.511	0.0294	−0.1075–0.1663	0.674
**ATC Class**	**J01A—Tetracyclines**	ARIMA (1,0,1)	−0.0080	−0.6181–0.6021	0.979	−0.0011	−0.0230–0.0208	0.921	(0,1,2) (1,0,0)[52]	−0.2091	−0.3789–−0.0393	**0.016**	−0.0062	−0.0131–0.0007	0.078
	**J01C—Beta-lactams**	(1,1,2) (0,0,1)[52]	−1.3452	−2.2672–−0.4233	**0.004**	0.0120	−0.0178–0.0418	0.432	(0,1,1) (0,1,1)[52]	−1.1940	−1.8611–−0.5269	**<0.001**	0.0091	−0.0164–0.0345	0.486
	**J01D—Other beta-lactams**	ARIMA (0,1,1)	−0.8461	−1.5165–−0.1756	**0.013**	0.0161	−0.0066–0.0387	0.165	(0,1,3) (1,0,0)[52]	−0.7622	−1.2316–−0.2927	**0.001**	0.0074	−0.0149–0.0297	0.514
	**J01E—Sulfonamides and trimethoprim**	ARIMA (0,0,0)	0.2489	−0.0894–0.5873	0.149	−0.0121	−0.0246–0.0003	0.055	(0,1,1) (0,0,1)[52]	0.2798	−0.0140–0.5737	0.062	0.0026	−0.0073–0.0124	0.607
	**J01F—Macrolides, lincosamides and streptogramins**	(1,0,1) (0,0,1)[52]	−0.2021	−0.5412–0.1369	0.243	0.0040	−0.0082–0.0161	0.520	(0,1,1) (1,0,0)[52]	0.4652	−0.2320–1.1623	0.191	−0.0196	−0.0622–0.0230	0.368
	**J01M—Quinolones**	(1,0,1) (0,0,1)[52]	−0.6517	−1.3175–0.0140	0.055	0.0038	−0.0201–0.0277	0.756	(0,1,1) (1,1,0)[52]	0.5290	−0.6442–1.7023	0.377	0.0032	−0.0605–0.0669	0.922
	**J01X—Other antibacterials**	ARIMA (0,0,2)	0.2889	−0.0695–0.6472	0.114	0.0005	−0.0125–0.0136	0.935	(1,0,1) (1,0,0)[52]	0.3660	−0.1521–0.8841	0.166	0.0107	−0.0061–0.0275	0.213
**Individual antibiotics**	**Amoxicillin**	(1,1,2) (0,0,1)[52]	−0.6484	−1.1083–−0.1886	**0.006**	0.0044	−0.0110–0.0198	0.577	(0,1,1) (1,0,0)[52]	−0.8057	−1.0684–−0.5430	**<0.001**	0.0126	0.0025–0.0228	**0.015**
	**Amoxicillin/clavulanic acid**	ARIMA (1,0,1)	−0.4003	−1.0344–0.2339	0.216	0.0045	−0.0184–0.0273	0.701	(0,1,1) (1,1,0)[52]	−0.0954	−0.7418–0.5511	0.773	−0.0142	−0.0442–0.0158	0.353
	**Azithromycin**	(1,0,1) (0,0,1)[52]	−0.2120	−0.5598–0.1358	0.232	0.0061	−0.0064–0.0186	0.340	(0,1,1) (0,1,1)[52]	0.7035	0.1592–1.2477	**0.011**	−0.0186	−0.0493–0.0121	0.234
	**Cephalexin**	ARIMA (0,1,1)	−0.6039	−1.1319–−0.0760	**0.025**	0.0107	−0.0067–0.0281	0.227	(2,0,2) (1,0,0)[52]	−0.3743	−0.7163–−0.0322	**0.032**	0.0115	−0.0005–0.0234	0.060
	**Clarithromycin ^#^**								(0,1,1) (1,0,0)[52]	−0.0225	−0.1376–0.0925	0.701	−0.0023	−0.0081–0.0035	0.433
	**Doxycycline**	ARIMA (1,1,1)	−0.2571	−0.9410–0.4268	0.461	−0.0001	−0.0305–0.0302	0.993	(1,1,1) (0,0,1)[52]	−0.2131	−0.3612–−0.0649	**0.005**	−0.0066	−0.0122–−0.0010	**0.020**
	**Fosfomycin ^#^**								(0,1,1) (1,0,0)[52]	0.0545	−0.2018–0.3109	0.677	−0.0063	−0.0162–0.0035	0.209
	**Nitrofurantoin**	ARIMA (2,0,2)	0.3159	−0.0001–0.6319	0.050	−0.0039	−0.0155–0.0076	0.506	(0,1,1) (1,0,0)[52]	−0.1981	−0.6301–0.2339	0.369	0.0099	−0.0088–0.0285	0.300
	**Penicillin ^#^**								ARIMA (0,1,1)	−0.0355	−0.0639–−0.0071	**0.014**	0.0009	−0.00009–0.0019	0.074

Bold: *p*-value < 0.05. * Model parameters are displayed as ‘SARIMA (p,d,q)(P,D,Q)S’. If no seasonality was present, the model is presented as: ARIMA (p,d,q). ^#^ No results are shown for clarithromycin, fosfomycin and penicillin in Alberta to preserve subject anonymity, as the number of dispensations was on average less than 5 per week. LTCF: Long-term care facility. SARIMA: seasonal autoregressive integrated moving average. ARIMA: autoregressive integrated moving average. 95% CI: 95% Confidence interval. ATC: Anatomical Therapeutic Chemical Classification. N.B. For amoxicillin and doxycycline in Ontario both a significant step and slope change was found. For amoxicillin, the slope change was positive while the step change was negative. This indicates that although there was a sudden decrease in prescribing rate detected directly after the start of the pandemic, this was accompanied by a minor increase in trend. For doxycycline a negative slope change was found in addition to a negative step change, indicating both a sudden decrease in prescription rates as well as a decreasing trend directly after the start of the pandemic.

## Data Availability

Restrictions apply to the availability of these data, which were obtained from administrative datasets that are not open source.

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
