# Peer review of "A Time Series Analysis Evaluating Antibiotic Prescription Rates in Long-Term Care during the COVID-19 Pandemic in Alberta and Ontario, Canada"

_antibiotics, 2022, doi:10.3390/antibiotics11081001_

Round 1

Reviewer 1 Report

I very much enjoyed reading this paper, which is well-written and covers an important topic. As the authors mention, very limited work has been done to evaluate how the COVID-19 pandemic has affected antibiotic prescribing for vulnerable groups such as LTCF residents. The study rationale is solid, and the methods are appropriate. My comments and suggestions are mainly aimed at improving clarity and facilitating interpretation of the study findings particularly for those who are unfamiliar with ITS analysis approaches. It may also be helpful to researchers willing to apply similar methods to other settings/populations. 

Major comments:

1) In the introduction, can you please explain the reasons behind the baseline difference in antibiotic prescription rates between Alberta and Ontario? Is a similar difference observed in the general population as well or is it LTCF-specific? Does it have to do with differences in approaches to antimicrobial stewardship in Alberta versus Ontario? 

2)    I would suggest clarifying whether LTCFs across Canadian provinces are public, private or a mix of those. I think this would help contextualize the study, particularly for those who are unfamiliar with the Canadian healthcare system.

3)  I strongly encourage the authors to remove p-values throughout the results section (particularly ITS analysis findings) and report 95% CIs instead. 

4)  From the description of ITS analysis results, it is unclear whether the reported estimates represent the change in level or the change in slope (e.g., lines 60-70). I am aware this is specified in the table, but I think it would be worth clarifying in text as well. Furthermore, it seems to me that the reported estimates refer to the change at the start of the pandemic (i.e. the timepoint indicated by the vertical line on the graph) relative to the previous week, adjusting for seasonal trends. For this reason, talking about change in prescription rates “during the pandemic period” may be misleading, so this point should be made clearer in the results. I know it is explained in the methods, but I think it is worth clarifying the wording in the results as well, particularly for those who are less familiar with ARIMA/SARIMA models.

5)  Discussion, lines 61-63: I think this sentence is unclear and would suggest rephrasing. Of note, which subgroups are you referring to? Also, can you clarify the meaning of “larger” in “…the impact of the COVID-19 pandemic was larger…”?

6)  Lines 166-167: Could you please elaborate more on this point? Also consider summarizing such “common indications for use” in the appendix.

7)   Can you please comment on the stationarity assumption for Ontario data? I wonder whether the time series was actually stationary given the progressively declining trend observed way before the start of the pandemic.

8)  I think it would be worth including one or two sentences discussing potential confounding. The models used accounted for seasonality and autocorrelation in the time series - perhaps you could note that the observation period (2017-2020) is short enough to assume that other factors could be ignored, if that is a reasonable assumption. 

9)   Consider including model equations somewhere in the manuscript or in the supplementary material. 

Minor comments:

1)  Table 1: I would suggest indicating the unit or type of measure being reported either at the top of each column or next to the variable name (not next to each category); in this regard, since both counts and percentages are shown, I would write “n (%)”. 

2)    Figure 1: Would it be possible to add some labels on the x-axis instead of just indicating the start of each year? Perhaps trimesters?

3)    Since Figure 3 is quite crowded, I wonder whether you could expand the y-axis to increase spacing between curves and hopefully make it easier to read.

4)    Consider adding AWaRe categories of antibiotics along with ATC codes.

5) Please add a reference for ATC classification (line 161) to clarify the version used. 

Author Response

Comments and Suggestions for Authors

I very much enjoyed reading this paper, which is well-written and covers an important topic. As the authors mention, very limited work has been done to evaluate how the COVID-19 pandemic has affected antibiotic prescribing for vulnerable groups such as LTCF residents. The study rationale is solid, and the methods are appropriate. My comments and suggestions are mainly aimed at improving clarity and facilitating interpretation of the study findings particularly for those who are unfamiliar with ITS analysis approaches. It may also be helpful to researchers willing to apply similar methods to other settings/populations.

We thank the reviewer for their positive feedback.

Major comments:

1) In the introduction, can you please explain the reasons behind the baseline difference in antibiotic prescription rates between Alberta and Ontario? Is a similar difference observed in the general population as well or is it LTCF-specific? Does it have to do with differences in approaches to antimicrobial stewardship in Alberta versus Ontario?

We acknowledge that there seems to be a (notable) difference in rates between the two provinces. When looking at the general population, the rates are comparable (see Crosby et al. CMAJ Open 2022, DOI:10.9778/cmajo.20210095). Exact numbers for LTCFs are not available, however. We believe the difference may be caused by differences in data handling, as we explain in the discussion:

“In addition, as data sources and database management are different between Alberta and Ontario; hence, rates may not be directly comparable between the provinces. For example, in Alberta there is no comprehensive list of LTCFs with internal pharmacies that do not submit data to the provincial dispensations database. Therefore, there is a possibility that the denominators are inflated.”

Since the focus of our paper is on trends and not on directly comparing rates, we chose not to highlight this difference in the introduction.

2)    I would suggest clarifying whether LTCFs across Canadian provinces are public, private or a mix of those. I think this would help contextualize the study, particularly for those who are unfamiliar with the Canadian healthcare system.

Thank you for the suggestion. A sentence was added to the Methods section to elaborate more on the structure in Canada: ‘Ownership of LTCFs in Canada can be either private (profit or non-for-profit) or public, and differences exist between the provinces [ref CIHI]. Data from all LTCFs in both provinces were included in this study.’

3)  I strongly encourage the authors to remove p-values throughout the results section (particularly ITS analysis findings) and report 95% CIs instead.

Thank you for the suggestion. The p-values in the results section are replaced by 95% Cis.

4)  From the description of ITS analysis results, it is unclear whether the reported estimates represent the change in level or the change in slope (e.g., lines 60-70). I am aware this is specified in the table, but I think it would be worth clarifying in text as well.

We’ve added the sentence ‘Reported changes in weekly prescription rates in this Results section refer to step changes.’ to the beginning of the Results section to clarify this point.

Furthermore, it seems to me that the reported estimates refer to the change at the start of the pandemic (i.e. the timepoint indicated by the vertical line on the graph) relative to the previous week, adjusting for seasonal trends. For this reason, talking about change in prescription rates “during the pandemic period” may be misleading, so this point should be made clearer in the results. I know it is explained in the methods, but I think it is worth clarifying the wording in the results as well, particularly for those who are less familiar with ARIMA/SARIMA models.

We have changed all occurrences of ‘during the pandemic period’ into ‘after the start of the pandemic’.

5)  Discussion, lines 61-63: I think this sentence is unclear and would suggest rephrasing. Of note, which subgroups are you referring to? Also, can you clarify the meaning of “larger” in “…the impact of the COVID-19 pandemic was larger…”?

What we meant to say is that we found significant changes in prescription rates in more ATC classes and more individual antibiotics in Ontario compared to Alberta. We’ve changed the sentence to: ‘Although we did not see a significant change in overall oral antibiotic prescription rates in Ontario contrary to what we found in Alberta, prescription rates of more ATC classes and individual antibiotics were affected in Ontario compared to Alberta.’

6)  Lines 166-167: Could you please elaborate more on this point? Also consider summarizing such “common indications for use” in the appendix.

We have rephrased and moved this sentence in the Methods section to clarify this point and a column is added to the Supplementary Table 1 containing the ‘most common indication for use’.

7)   Can you please comment on the stationarity assumption for Ontario data? I wonder whether the time series was actually stationary given the progressively declining trend observed way before the start of the pandemic.

There was indeed a clear declining trend observed in the Ontario data. However, differencing was applied to induce stationarity. No limit was set to the value of d and D in the models and the data seemed sufficiently stationary after transformation.

8)  I think it would be worth including one or two sentences discussing potential confounding. The models used accounted for seasonality and autocorrelation in the time series - perhaps you could note that the observation period (2017-2020) is short enough to assume that other factors could be ignored, if that is a reasonable assumption.

Thank you for the suggestion, that is a valuable addition. The following was added to the Discussion: ‘Besides accounting for seasonality and autocorrelation, we did not take other confounders into account in our models. However, since our study period is relatively short, we assume that there will not have been relevant changes in other important factors in this time period.’

9)   Consider including model equations somewhere in the manuscript or in the supplementary material.

Thank you for the suggestion. We have considered this, but we believe this might be too technical for the intended public. All model assumptions and parameters (and R-packages used) are described in the methods section, which should allow other researchers to apply similar methods to their own data.

Minor comments:

1)  Table 1: I would suggest indicating the unit or type of measure being reported either at the top of each column or next to the variable name (not next to each category); in this regard, since both counts and percentages are shown, I would write “n (%)”.

The Table is adapted accordingly.

2)    Figure 1: Would it be possible to add some labels on the x-axis instead of just indicating the start of each year? Perhaps trimesters?

Thank you for the suggestion. Trimesters are added to the Figures.

3)    Since Figure 3 is quite crowded, I wonder whether you could expand the y-axis to increase spacing between curves and hopefully make it easier to read.

We agree. We’ve changed the y-axis, however, especially for Alberta the rates are overlapping on many points, thereby still causing a bit of an unclear picture. If the paper is accepted, we will discuss with the journal editorial team if it needs to be changed further to be more clear in the final publication.

4)    Consider adding AWaRe categories of antibiotics along with ATC codes.

Thank you for the suggestion. We’ve added this to Supplementary Table 1.

5) Please add a reference for ATC classification (line 161) to clarify the version used.

Apologies, the references were mixed up when the paper was restructured. The correct reference is now cited here.

Reviewer 2 Report

The reduction of antibiotics for systemic use is well known and reported several times in the literature (even from Canada). Blix HS, Høye S. Use of antibiotics during the COVID-19 pandemic. Tidsskr Nor Laegeforen. 2021 Feb 12;141(4). English, Norwegian. doi: 10.4045/tidsskr.20.1003. PMID: 33685110.Gillies MB, Burgner DP, Ivancic L, Nassar N, Miller JE, Sullivan SG, Todd IMF, Pearson SA, Schaffer AL, Zoega H. Changes in antibiotic prescribing following COVID-19 restrictions: Lessons for post-pandemic antibiotic stewardship. Br J Clin Pharmacol. 2022 Mar;88(3):1143-1151. doi: 10.1111/bcp.15000. Epub 2021 Aug 17. PMID: 34405427; PMCID: PMC8444718. Knight BD, Shurgold J, Smith G, MacFadden DR, Schwartz KL, Daneman N, Gravel Tropper D, Brooks J. The impact of COVID-19 on community antibiotic use in Canada: an ecological study. Clin Microbiol Infect. 2022 Mar;28(3):426-432. doi: 10.1016/j.cmi.2021.10.013. Epub 2021 Oct 30. PMID: 34757115; PMCID: PMC8556063.

It is an ecological study comparing the antibiotic consumption in long-term care facilities in two major areas of Canada. among elderly inhabitants. It might be considered as an additional report of the reduced antibitic consumption during Covid-19 pandemic.

It could be published without any modification.

Author Response

The reduction of antibiotics for systemic use is well known and reported several times in the literature (even from Canada). Blix HS, Høye S. Use of antibiotics during the COVID-19 pandemic. Tidsskr Nor Laegeforen. 2021 Feb 12;141(4). English, Norwegian. doi: 10.4045/tidsskr.20.1003. PMID: 33685110.Gillies MB, Burgner DP, Ivancic L, Nassar N, Miller JE, Sullivan SG, Todd IMF, Pearson SA, Schaffer AL, Zoega H. Changes in antibiotic prescribing following COVID-19 restrictions: Lessons for post-pandemic antibiotic stewardship. Br J Clin Pharmacol. 2022 Mar;88(3):1143-1151. doi: 10.1111/bcp.15000. Epub 2021 Aug 17. PMID: 34405427; PMCID: PMC8444718. Knight BD, Shurgold J, Smith G, MacFadden DR, Schwartz KL, Daneman N, Gravel Tropper D, Brooks J. The impact of COVID-19 on community antibiotic use in Canada: an ecological study. Clin Microbiol Infect. 2022 Mar;28(3):426-432. doi: 10.1016/j.cmi.2021.10.013. Epub 2021 Oct 30. PMID: 34757115; PMCID: PMC8556063.

It is an ecological study comparing the antibiotic consumption in long-term care facilities in two major areas of Canada. among elderly inhabitants. It might be considered as an additional report of the reduced antibitic consumption during Covid-19 pandemic.

It could be published without any modification.

Thank you for the positive feedback.

Although there are already papers published on (reduced) antibiotic consumption during the COVID-19 pandemic, literature on LTCF residents is scarce. Therefore we believe this paper adds valuable information regarding this population.

Reviewer 3 Report

Overall, this paper on the evaluating antibiotic prescription rates in long-term care facilities during the COVID-19 pandemic in Alberta and Ontario, Canada is well written and addresses an important topic. Below are a few recommended changes to improve the manuscript.

General:

-There are some citations that do not seem to match up with what you are saying in the text. Examples include line 45 (citation 13) and line 54 (citation 14). However please go back and double check ALL citations.

-You mention not being able to add 2021 data. What is the reason?

Figure 1, 2, and 3: The writing in these images is difficult to see. In addition, figure 3 is a little hard to see as well.  Perhaps it would be clearer after publication.

Results:

-Line 103-104: Not sure I'd classify amox/clav as a drug used to treat SSTI

Discussion:

-line 48-49: You mention that study found a significant increase in doxycycline prescribing, wheras I believe your study found a decrease. How is this "in line" with your results?

-Line 52-54: Does this study utilize all the same nursing homes in Ontario? If so, what is different between the studies outside of just also collecting IV? Did that study split it up so you could see the oral results by itself? If so, what would this study add? Could be worth commenting. However, I was not able to double check since the citation was incorrect.

-Line 62-63: Is this true? I thought it was the opposite (no significant change in Ontario).

Author Response

Comments and Suggestions for Authors

Overall, this paper on the evaluating antibiotic prescription rates in long-term care facilities during the COVID-19 pandemic in Alberta and Ontario, Canada is well written and addresses an important topic. Below are a few recommended changes to improve the manuscript.

Thank you for the positive feedback.

General:

-There are some citations that do not seem to match up with what you are saying in the text. Examples include line 45 (citation 13) and line 54 (citation 14). However please go back and double check ALL citations.

Apologies, when restructuring the paper, the references were mixed up. We’ve corrected this.

-You mention not being able to add 2021 data. What is the reason?

The project was initially designed to look at the time period 2017-2020 and the provinces provided data for this time period. Although the final paper was finished in 2022, unfortunately we did not have access to data after 2020.

Figure 1, 2, and 3: The writing in these images is difficult to see. In addition, figure 3 is a little hard to see as well.  Perhaps it would be clearer after publication.

Thank you, we agree. We’ve enlarged the writing and changed some formatting. If the paper is accepted, we will discuss with the journal editorial team if it needs to be changed further to be more clear in the final publication.

Results:

-Line 103-104: Not sure I'd classify amox/clav as a drug used to treat SSTI

We agree, it is a broad spectrum antibiotic also used for SSTI, but mainly indicated for RTI. We’ve deleted this sentence from the manuscript.

Discussion:

-line 48-49: You mention that study found a significant increase in doxycycline prescribing, wheras I believe your study found a decrease. How is this "in line" with your results?

We agree, this is not in line with our results. This discrepancy is now mentioned in the discussion.

-Line 52-54: Does this study utilize all the same nursing homes in Ontario? If so, what is different between the studies outside of just also collecting IV? Did that study split it up so you could see the oral results by itself? If so, what would this study add? Could be worth commenting. However, I was not able to double check since the citation was incorrect.

The study mentioned did use the same nursing homes in Ontario, but they only looked at overall antibiotic use, including both oral and non-oral drugs. Their results were not stratified by oral vs. other routes of administration. Also, no analyses were done on the different ATC classes and individual antibiotics. We’ve added a clarification to this sentence in the Discussion.

-Line 62-63: Is this true? I thought it was the opposite (no significant change in Ontario).

Indeed, there was no significant change on overall prescription rates in Ontario. However, what we meant to say is that we found significant changes in prescription rates in more ATC classes and more individual antibiotics in Ontario compared to Alberta. We’ve changed the sentence to: ‘Although we did not see a significant change in overall oral antibiotic prescription rates in Ontario contrary to what we found in Alberta, prescription rates of more ATC classes and individual antibiotics were affected in Ontario compared to Alberta.’